# Clinical and Epidemiological Characteristics of Pediatric Pertussis Cases: A Retrospective Study from Southeast Romania

**DOI:** 10.3390/antibiotics14050428

**Published:** 2025-04-23

**Authors:** Cristina Maria Mihai, Ancuta Lupu, Tatiana Chisnoiu, Adriana Luminita Balasa, Ginel Baciu, Silvia Fotea, Vasile Valeriu Lupu, Violeta Popovici, Simona Claudia Cambrea, Mircea Grigorian, Felicia Suciu, Florin-Daniel Enache, Anna Sora, Ramona Mihaela Stoicescu

**Affiliations:** 1Pediatric Department, Faculty of General Medicine, “Ovidius” University, 900470 Constanta, Romania; cristina_mihai@365.univ-ovidius.ro (C.M.M.); adriana.balasa@365.univ-ovidius.ro (A.L.B.); 2Pediatrics, County Clinical Emergency Hospital of Constanta, 900591 Constanta, Romania; florin.enache@365.univ-ovidius.ro; 3Pediatric Department, “Grigore T. Popa” University of Medicine and Pharmacy, 700115 Iasi, Romania; ancuta.ignat1@umfiasi.ro (A.L.); vasile.lupu@umfiasi.ro (V.V.L.); 4Pediatric Department, “Dunărea de Jos” University of Galati, 800008 Galati, Romania; baciuginel@ugal.ro (G.B.); foteasilvia@ugal.ro (S.F.); 5Center for Mountain Economics, “Costin C. Kritescu” National Institute of Economic Research (INCE-CEMONT), Romanian Academy, 725700 Vatra-Dornei, Romania; 6Department of Infectious Diseases, Faculty of General Medicine, “Ovidius” University, 900470 Constanta, Romania; claudia.cambrea@365.univ-ovidius.ro; 7Department of Physiology and Physiopathology, Faculty of Dental Medicine, “Ovidius” University, 900178 Constanța, Romania; mirceagrigorian@yahoo.com; 8Department of Analysis and Quality Control of Drugs, Faculty of Pharmacy, “Ovidius” University, 900470 Constanta, Romania; felicia.suciu@365.univ-ovidius.ro; 9Department of Pediatric Surgery and Orthopedics, Faculty of General Medicine, “Ovidius” University of Constanta, 900470 Constanta, Romania; 10Center for Research and Development of the Morphological and Genetic Studies of Malignant Pathology (CEDMOG), Blvd. Tomis nr. 145, 900591 Constanta, Romania; alexiu_anna@yahoo.com; 11Department of Microbiology and Immunology, Faculty of Pharmacy, “Ovidius” University of Constanta, Str. Căpitan Aviator Al. Șerbănescu, nr.6, Campus Corp C, 900470 Constanta, Romania; ramona.stoicescu@univ-ovidius.ro

**Keywords:** *Bordetella pertussis*, whooping cough, pediatric patients, *B. pertussis* vaccine, co-infections, hyperleukocytosis, macrolides

## Abstract

**Background/Objectives:** Pertussis remains a significant cause of respiratory illness in children, particularly in regions with suboptimal vaccination coverage. This retrospective study analyzes the clinical presentations, co-infections, treatment, and outcomes of pediatric patients diagnosed with *Bordetella pertussis* at the Constanța County Clinical Emergency Hospital “St. Apostle Andrew” between 1 January and 30 September 2024. **Methods:** Thirty-eight children, predominantly under the age of 3 years (81.58%), were included. Demographic data, clinical features, coinfecting pathogens, antimicrobial regimens, and hospital outcomes were reviewed. **Results:** Only 7 out of 38 children (18.42%) had received pertussis vaccination, and none benefited from maternal immunization. The highest incidence occurred in infants under 1 year (44.74%). Intensive care was required in 18.42% of cases, and macrolides were the most frequently used antibiotics (68.42%). Co-detection of respiratory pathogens—particularly *Streptococcus pneumoniae*, enteroviruses, and human rhinoviruses—was common. Severe cases often exhibited hyperleukocytosis, which was associated with complications such as heart failure. **Conclusions:** These findings underscore the need for timely recognition and management of pertussis and its complications. Although macrolides remain the first-line therapy, adjunctive treatments like leukoreduction may be considered in critical cases. The persistence of pertussis despite vaccination efforts highlights the challenges posed by waning immunity and diagnostic limitations, reinforcing the need for strengthened public health strategies.

## 1. Introduction

Pertussis, also known as whooping cough, is a respiratory infection caused by *Bordetella pertussis* [1]. The hallmark of classic pertussis is a prolonged cough that can persist for several weeks. The illness is characterized by intense bouts of coughing, known as paroxysms, which often conclude with a distinctive gasping sound called a “whoop” [2].

Pertussis is a vaccine-preventable disease, and introducing the pertussis vaccine in the 20th century has significantly reduced the number of pertussis cases and deaths in children [3]. Pertussis continues to be a primary public health concern, with significant rates of illness and death [4]. According to the United States Centers for Disease Control and Prevention (CDC), about one-third of infants with pertussis require hospitalization, and 1% of these hospitalized cases result in death [5]. Pertussis remains one of the leading causes of illness and mortality among infants [6]. Whooping cough affects individuals of all ages, but it is most severe in infants, who have the highest age-specific incidence of the disease and account for nearly all hospitalizations and deaths related to pertussis [7].

Globally, pertussis affects millions of individuals annually, with children bearing the brunt of the disease burden [8]. The World Health Organization estimates that pertussis causes over 160.000 deaths annually, primarily in infants under 6 months of age who are not fully vaccinated [9]. Severe complications, including pneumonia, encephalopathy, and even death, underscore the disease’s high stakes in pediatric health. Such outcomes are particularly concerning in resource-limited settings, where access to vaccines, diagnostic tools, and medical care may be constrained [10].

Despite decades of extensive immunization campaigns, whooping cough seriously threatens world health [11]. This highly infectious disease can cause severe respiratory difficulties, especially in newborns and young children. Periodic illness outbreaks in different communities highlight the intricacy of its epidemiology and the shortcomings of existing prevention interventions, even though vaccination programs have significantly decreased pertussis-related morbidity and death [12].

Even though acellular vaccinations (aP) have fewer side effects than whole-cell vaccines (wP), concerns have been raised over their long-term effectiveness and the necessity of booster shots due to their shorter lifetime of protection [13]. Furthermore, discussions about vaccination methods, such as the timing and makeup of immunization regimens, have been triggered by the impact of the pertussis comeback in highly vaccinated populations [14].

The persistence of pertussis as a public health threat is multifaceted [15,16]. Waning immunity after vaccination or natural infection, incomplete vaccine coverage, and the emergence of *Bordetella pertussis* strains with reduced vaccine sensitivity contribute to the ongoing burden of disease [17]. Moreover, diagnostic challenges, including nonspecific early symptoms and limited access to advanced diagnostic tools, can delay treatment and containment, exacerbating its spread [18]. For this reason, pertussis is also known as “the 100-day cough.” [19]. These factors highlight the need for continued research to refine preventive and management strategies for pertussis, especially in pediatric populations.

Pertussis typically begins with symptoms resembling a mild upper respiratory infection [20]. A mild cough progressively intensifies within 1 to 2 weeks, becoming paroxysmal. These coughing fits increase in frequency and severity before gradually subsiding over several weeks or longer [21]. Paroxysms are characterized by rapid coughs without inhalation, followed by the characteristic “whoop,” which is a desperate effort to breathe through a swollen glottis [22]. During a paroxysm, the patient may turn cyanotic, and vomiting often follows the intense coughing. Multiple paroxysms can occur in quick succession, leaving the patient exhausted. Moreover, they tend to be more severe at night. Between these episodes, the patient generally appears normal [23]. While fever is uncommon in pertussis, the condition is often associated with elevated levels of neutrophils, C-reactive protein (CRP), and procalcitonin, indicating the possible presence of concurrent bacterial infections [24]. Even after the illness resolves, the cough can persist for many weeks, and intercurrent viral infections may provoke a recurrence of the paroxysms [25,26,27].

Leukocytosis is a hallmark of pertussis [28]. Studies have shown that children with severe pertussis exhibit significantly higher white blood cell counts than those with milder forms of the disease, with particularly elevated levels observed in fatal cases [29]. The poor deformability of these white blood cells leads to obstruction in the narrowed alveolar capillary beds, resulting in embolism due to the formation of leukocyte clumps [30]. This embolism causes hypoxemia and pulmonary hypertension, which can impair cardiac function and lead to heart failure in severe cases [31,32]. Severe hyperleukocytosis is recognized as an independent risk factor for malignant pertussis, a life-threatening form of the disease [33].

There has been a rise in the incidence of whooping cough in the EU/EEA. In Romania, 112 cases of whooping cough were recorded in the first 4 months of 2024. It marks a significant increase compared to only 16 cases reported in 2023; it is also higher than the average of 93 cases reported annually during the 5 pre-pandemic years (2015–2019) [34]. The cases belong to 22 Romanian counties, with the most affected age group being 0–4 years. Of these cases, 33% were infants, of whom 89% were eligible for vaccination; the remaining 11% were too young to receive the pertussis vaccine, as they were under 2 months of age [35]. Two months later, this number dramatically increased to 417; two deaths were recorded, both in infants (in Bucharest and Iasi County) [36].

In this context, the present study aims to fill critical gaps in understanding pertussis’s clinical and epidemiological aspects in pediatric populations. By integrating detailed analyses of patient presentations, disease outcomes, and vaccination statuses alongside a comprehensive literature review, this work seeks to (i) highlight the complex interplay between vaccination, immunity, and disease resurgence, (ii) explore the unique vulnerabilities of children to pertussis, including age-specific risks and outcomes, and (iii) address diagnostic and management challenges, particularly in resource-limited contexts.

## 2. Results

### 2.1. Socio-Demographic and Baseline Data of the Pediatric Patients

The retrospective study was performed on 38 pediatric patients, 19/38 (50%) boys and 19/38 (50%) girls, 17/38 (44.74%) with rural residence, and 21/38 (55.26%) from urban zones. Statistically significant differences are reported between boys and girls of age groups <1–3 and 4–13 years, *B. pertussis* vaccination, and clinical laboratory diagnosis method (*p* < 0.05). The data are recorded in Table 1.

Most pediatric patients (31/38, 81.58%) have up to 3 years, while only 7/38 (18.42%) have 9–13 years (*p* < 0.05, Table 1). Only 7/38 (18.42%) received at least one dose of *B. pertussis* vaccine, while 31/38 (81.58%) were not vaccinated (*p* < 0.05). No pediatric patients were protected through a maternal vaccine. Around 32/38 (84.21%) were diagnosed through real-time PCR (qPCR), *p* < 0.05 (Table 1).

Serological diagnosis (ELISA) of *B. pertussis* was performed for 6/38 (15.79%) patients (Table 1). These patients showed elevated anti-*B. pertussis* IgG antibody titers, exceeding the diagnostic threshold of >100 IU/mL. These serologically confirmed cases were clinically consistent with pertussis and included in the analysis. The diagnostic approach was aligned with the clinical signs, timing of sample collection, and immunization history.

IgM and IgA levels were not assessed, as these assays were not part of the standard diagnostic protocol at our institution during the study period.

The correlation matrix from Appendix A and Figure 1 supports the data from Table 1, evidencing a substantial correlation between urban residence and the 1–3 years age group and female pediatric patients and the 4–8 years age group (r = 0.999, *p* < 0.05). BPV Yes strongly correlates with age group 9–13 years and RT-PCR diagnosis (r = 0.961, r = 0.915, *p* > 0.05). The rural zone is moderately associated with boys and urban residence with girls (r = 0.731, r = 0.693, *p* > 0.05).

### 2.2. Associated Pathogens and Clinical Manifestations

*B. pertussis* was associated with several viral pathogens in pediatric patients. Enterovirus-human-rhinovirus (EV-HRV) was detected in 12/38 patients (31.58%), Coronavirus SARS-CoV-2 in 5/38 (13.16%), and Human parainfluenza virus type 3 (HPIV-3) in 2/38 (5.26%), *p* < 0.05 (Table 2). Human adenovirus (HAdVs), Measles virus (MV), and respiratory syncytial virus (RSV) were each detected in 1/38 (2.63%) pediatric patients. Other identified pathogens in pediatric patients were *Streptococcus pneumoniae* (7/38 patients, 18.42%), *Mycoplasma pneumoniae*, and *Pneumocystis jirovecii* (1/38 patients, 2.63%), *p* < 0.05 (Table 2).

Around 36/38 patients (94.74%) claimed respiratory symptoms several days before presentation in the ECU (RSB), while only 2/38 (5.26%) did not have them, *p* < 0.05. A total of 24/38 patients (63.16%) manifested respiratory symptoms 4–14 days before, while 7/38 (18.42%) and 5/38 (13.16%) claimed them 1–3 days and over 20 days before admission, *p* < 0.05. Most pathogens were co-detected in pediatric patients with respiratory symptoms 4–9 days before presentation in the ECU (12/38, 31.58%). Thus, *S. pneumoniae* was co-detected in 6/12 patients (50%), EV-HRV and SARS-CoV-2 were each co-detected in 3/12 patients (25%), and HPIV-3, HAdVs, *M. pneumoniae,* and *P. jirovecii* were each co-detected in 1/12 (8.33%) patients. EV-HRV and MV were co-detected in 4/5 children (80%) with respiratory symptoms 20 days before admission to the pediatric department. Other pathogens (HPIV-3, SARS-CoV-2, RSV, and *S. pneumoniae*) were co-detected in 5/7 (71.43%) patients with respiratory syndrome 1–3 days before, and EV-HRV was co-detected in only 3/12 (25%) patients with 10–14 days (Table 2).

Pearson correlation reports a substantial correlation between RSB > 20 and MV, RSB4-9 and HAdVS, *S. pneumoniae*, *M. pneumoniae*, and *P. jirovecii,* as well as RSB1-3 and RSV (r = 0.986–0.999, *p* < 0.05).

Cough was the main symptom identified in all pediatric patients (38/38, 100%). A total of 16/38 patients (42.11%) had a fever; high fever (>39 °C) was detected in 3/38 patients (7.89%), *p* < 0.05. Around 14/38 (36.84%) had rhinorrhea, and 7/38 (18.42%) had oxygen desaturation (O2 DS), *p* < 0.05. Other symptoms include emesis (7/38, 18.42%) and rash (1/38, 2.63%), *p* < 0.05.

Pearson Correlation shows that RSB > 20 significantly correlates with rash and RSB4-9 with emesis (r = 0.999, r = 0.962, *p* < 0.05), and rhinorrhea is strongly associated with O2 DS and fever (r = 0.971–0.997, *p* < 0.05). Morbillivirus (MV) substantially correlates with rash and emesis with HAdVs, *S*. *pneumoniae, M. pneumoniae,* and *P. jirovencii*; rhinorrhea and O2 DS strongly correlate with SARS-CoV2 and HPIV-3, as well as fever with SARS-CoV-2 (r = 0.962–0.999, *p* < 0.05).

Among the 38 pediatric patients, 23 (60.53%) had signs of lower respiratory tract infection (LRTI), including bronchitis and pneumonia. Two patients developed acute respiratory distress syndrome (ARDS), as defined by the Berlin criteria [37]. They required intensive respiratory support, including mechanical ventilation. Due to the severity of ARDS and clinical implications, these cases were considered separately and were not classified under general LRTIs.

### 2.3. Clinical Laboratory and Radiological Investigations

The corresponding data are registered in Table 3.

Eighteen children (18/38, 47.37%) had WBCs in the normal range, while the same number had increased WBCs (20–100 × 10^3^). Only 2/38 (5.26%) had WBCs > 100 × 10^3^. Both deceased patients had WBCs > 20 × 10^3^; the other 9/38 children (74%) were transferred to another healthcare unit (*p* < 0.05, Table 3). Of the 8/38 pediatric patients with Lym cells ≤ 40%, 2/38 were deceased, and one was transferred (*p* < 0.05, Table 3). PBS indicated significant changes in blood cells in 7/38 children (18.42%); a total of 1/7 were deceased, and 5/7 were transferred (*p* < 0.05). Only 1/7 remained hospitalized for clinical cure.

Six patients (6/38, 15.79%) had substantially high CRP levels (11—>100 mg/dL); a total of 2/6 (33%) were deceased, and 2/6 (33%) were transferred during the first 8 days of hospitalization (Table 3).

Thirty patients (30/38, 78.95%) had positive Chest X-ray exams; seven different results were obtained, noting C-X-ray 1–7: C-X-ray 1—Accentuated interstitial pattern below the hilum bilaterally; C-X-ray 2—Alveolar opacities around and below the right hilum; C-X-ray 3—Bilateral pulmonary infiltrate; C-X-ray 4—Blurring of the right basal pulmonary field: C-X-ray 5—Congestive pulmonary hila, microalveolar and reticular opacities around and below the hilum bilaterally; C-X-ray 6—Enlarged congestive hila, confluent alveolar opacities around the left hilum and below the hilum bilaterally; C-X-ray 7—Right pulmonary consolidation process; C-X-ray 8—Widespread, homogeneous opacities of medium intensity, with blurred margins, showing air bronchograms, located in the upper third of both lung fields and left retrocardiac area—indicative of pulmonary infiltrates. All results of the Chest X-ray exams are displayed in Figure 2, distributed in the whole group of pediatric patients (Figure 2A), and associated with clinical outcomes (Figure 2B,D).

Around 23/38 (60.53%) had a bilaterally accentuated interstitial pattern below the hilum (C-X-ray 1); similar numbers (1/38, 2.63%) had C-X-ray 2–8. Four children (4/38, 10.53%) had no Chest X-ray exam, and the other 4/38 had a normal chest image (Figure 2A).

The patients with C-X-ray 1 (62.50%), 2 (12.50%), and 7 (12.50%) had clinical cures; those with C-X-ray 1 (66.67%), 4, 5, and 6 (each 8.33%) were transferred; the deceased patients had C-X-ray 3 and 8 (Figure 2B–D).

Four children had complications: Bilateral apical-lateral-basal pneumothorax; pulmonary condensation process in the lower one-third of the right pulmonary field; confluent alveolar opacities; interstitial peri- and hylo-basal bilateral enlarged congestive hilar regions, aspiration pneumonia, kidney failure, paroxysmal manifestations, and anasarca. Two patients died, and one was transferred (Table 3). Sixteen patients (16/38, 42.11%) had HD = 6–10 days, and 12/38 (31.58%) were moved in the first 1–8 days (*p* < 0.05). A total of 2/38 (5.26%) had >10 HD, 4/38 (10.53%) had 2–5 days, and the other 4/38 were not hospitalized.

The correlation matrix from Appendix A and Figure 3 were used to analyze the correlations between the variable data from Table 3. They show that Lym < 20% substantially correlates with C-Yes, Lym > 40% with C-X-ray positive, PBS positive with CRP > 100 mg/dL, PBS No with C-X-Ray normal, and C-X-ray no with HD No and HD = 3–5 (r = 0.998—0.999, *p* < 0.05). C-No highly correlates with HD No and HD = 3.5 (r = 0.878, *p* > 0.05), and CRP > 100mg/dL significantly correlates with transfer after 1–8 days (r = 0.999, *p* < 0.05).

### 2.4. Treatment and Evolution

Antimicrobial treatment involves antibacterials and antivirals. The antibacterial treatment administered in hospitalized pediatric patients includes macrolides (Azithromycin and Clarithromycin) and beta-lactams (Amoxicillin and Ceftriaxone). The administration route, dosage per dose and day, and treatment duration differ by antibiotic type. Macrolides were administered orally (Azithromycin (10 mg/kg, maximum 500 mg, 3- 5 days) and Clarithromycin (7.5 mg/kg/dose twice daily (maximum 500 mg per dose)). From beta-lactams, Ceftriaxone was administered intravenously (IV) at 50–100 mg/kg/day (maximum 2 g/day) in one or two divided doses at 7–10 days. Amoxicillin was administered orally (20–40 mg/kg/day in two or three divided doses (maximum 500 mg/dose)).

The antiviral therapy was used for co-infections with other viral pathogens, such as Remdesivir for cases involving SARS-CoV-2 or other respiratory viruses, based on clinical indication: Remdesivir IV (loading dose: 5 mg/kg on day 1 (maximum 200 mg); maintenance dose: 2.5 mg/kg/day (maximum 100 mg) from day 2, 5–10 days, depending on the clinical response).

In the present study, 26/38 (68.42%) patients received macrolides: Azithromycin (21/38, 55.26%) and Clarithromycin (5/38, 13.16%), *p* < 0.05. A total of 10/38 patients were treated with beta-lactams: Ceftriaxone (8/38, 21.05%) and Amoxicillin (2/38, 5.26%), *p* < 0.05 (Table 4). Two children had Remdesivir (2/38, 5.26%).

Of the pediatric patients, 7/38 (18.42%) needed intensive care in the pediatric intensive care unit (PICU, Table 4).

In the studied pediatric group, 37/38 patients received anti-inflammatory steroids: Dexamethasone (Dex, 21/38, 55.26%) and Hydrocortisone hemisuccinate (HHC, 14/38, 36.84%), and 31/38 (81.58%) patients received nebulized adrenaline (*p* < 0.05, Table 4). Furosemide was administered to 2/38 children (5.26%), while only 1/38 (2.63%) received Aminophylline, Epinephrine, Norepinephrine, Mannitol, and Albumin.

Oxygen therapy was necessary for 22/38 patients: Low-flow nasal cannula O2 therapy (LFNCO2, 18/38, 47.37%) and High-flow nasal cannula O2 therapy (HFNCO2, 4/38, 10,53%), *p* < 0.05. For 3/38 children (7.89%), nasogastric intubation was necessary, with 2/38 (5.26%) receiving mechanical ventilation and blood transfusion, and 1/38 (2.63%) were supposed to have bilateral pleurotomy (*p* < 0.05, Table 4).

Principal Component Analysis was used to investigate the correlations between the variable parameters registered in Table 4. The correlation matrix from Appendix A and Figure 4 show that intensive care in the PICU is substantially associated with Albumin, Aminophylline, Mannitol, Epinephrine, Clarithromycin, and clinical cure (r = 0.999, *p* < 0.05) and strongly correlates with LFNCO2, HFNCO2, nasogastric intubation, Azithromycin, and Dex (r = 0.899–0.990, *p* > 0.05).

Transfer significantly correlates with bilateral pleurotomy, blood transfusion, and Norepinephrine (r = 0.999, *p* < 0.05) and is strongly associated with HHC and Ceftriaxone (r = 0.929–0.945, *p* > 0.05). Mechanical ventilation is remarkably correlated with Furosemide, Remdesivir, and Amoxicillin (r = 0.999, *p* < 0.05) and shows a good correlation with nasogastric intubation and nebulized adrenaline (r = 0.849–0.866, *p* > 0.05). Bilateral pneumotomy strongly correlates with Ceftriaxone and HHC (r = 0.929–0.945, *p* > 0.05).

### 2.5. Correlations of Socio-Demographic and Baseline Factors and Clinical Findings

Our study shows that pertussis is diagnosed more frequently in urban regions (21/38, 55.26%) than in rural zones (17/38, 44.74%), *p* < 0.05 (Table 5).

In addition, almost all vaccinated pediatric patients had urban residences (85.71% vs. 14.29%, *p* < 0.05). Of the 38 pediatric patients diagnosed with Bordetella pertussis infection, the majority (81.58%) were aged 3 years or younger, and 55.26% lived in urban areas. Only 7 children (18.42%) had a documented history of pertussis vaccination; four had incomplete vaccination schedules despite initial parental reports of full immunization. The clinical outcomes, hospitalization period, and therapeutic protocol analysis highlight the benefits of pertussis vaccination in pediatric patients (Table 5). Hence, the incidence of fever, LRTI, rhinorrhea, and O2 DS is appreciably lower in vaccinated children (28.57% vs. 45.16% and 67.74%; 14.29% vs. 41.94%; and 0% vs. 22.58%, *p* < 0.05). The highest WBC, Lym, and CRP levels were measured in non-vaccinated pediatric patients (6.45% vs. 0%; 35.48% vs. 14.29%; and 3.23% vs. 0%, *p* < 0.05). Only 57.14% of BPV-Yes patients were hospitalized vs. 96.77% of the non-vaccinated patients, *p* < 0.05 (Table 5). Vaccinated children had no complications (100% vs. 87.10%, *p* < 0.05), and they did not need HFNCO2 and special care in the PICU (0% vs. 12.90% and 22.58%, respectively, *p* < 0.05). The deceased children were unvaccinated (6.45% vs. 0%, *p* < 0.05).

Principal Component Analysis supports and details the previously mentioned results. Thus, the correlation matrix from Appendix A and Figure 5 indicate that complications C1–C3 are substantially associated with death (r = 0.999, *p* < 0.05). BPV No is significantly associated with rural residence, RSB = 1–3 days and 4–9 days, HD Yes, and PICU intensive care (r = 0.957–0.975, *p* < 0.05). It also shows a substantial correlation with complications C1–C4 and death prognostic (r = 0.808, *p* > 0.05), RSB = 10–14 days, and transfer (r = 0.785–0.761, *p* > 0.05).

Both rural and urban residences are substantially correlated with HD Yes and clinical cure (r = 0.973–0.996, *p* < 0.05), and they are moderately associated with complications (C1–C3), r = 0.757–0.662, *p* > 0.05 (Figure 5).

C1–C3 are moderately associated with RSB > 20 days, RSB = 4–9 days, and clinical cure (r = 0.730–0.786, *p* > 0.05), and they evidence a high correlation with PICU intensive care (r = 0.917, *p* > 0.05). Moreover, Figure 5 also indicates the place of each age group associated with all baseline and clinical outcomes: infants and young children up to 3 years are the most vulnerable pediatric patients.

Two patients in our cohort, both under 3 months of age and unvaccinated, experienced severe disease progression and, unfortunately, died. Both developed acute respiratory distress syndrome (ARDS), requiring mechanical ventilation and intensive supportive care. One patient presented with a white blood cell count exceeding 80,000/µL, raising concern for leukostasis-related complications. Despite aggressive treatment, including macrolide antibiotics and respiratory support, both children succumbed to cardiorespiratory failure within 72 h of admission. No underlying immunodeficiencies were identified.

## 3. Materials and Methods

A retrospective clinical study was performed on 38 pediatric patients diagnosed with *Bordetella pertussis* infection from Southeast Romania. The study was approved by the Ethical Committee of Constanta County Clinical Emergency Hospital “St. Apostle Andrew,” Document number 08, approved on 5 March 2025. The pediatric patients were children aged < 1 year and 13 years, all of whom were hospitalized with pertussis in the Pediatric Departments of the hospital between 1 January 2024 and 30 September 2024.

### 3.1. Data Collection

Data were extracted from hospital databases and medical charts, including demographic details (age and gender), clinical features, length of hospital stay, and complications. Clinical evaluation, laboratory tests, and imaging techniques confirmed the diagnosis of *B. pertussis* infection.

This retrospective cohort study includes all pediatric patients diagnosed and treated as inpatients with pertussis at our institution during the observation period from 1 January 2024 to 30 September 2024. The inclusion criteria were as follows:Age range: Hospitalized pediatric patients aged less than 1 year to 13 years;Diagnosis of pertussis: All patients included in the study had a confirmed diagnosis of pertussis, which was determined through clinical evaluation and positive molecular testing for Bordetella pertussis (PCR);Inpatient admission: Only patients hospitalized in the Pediatric Departments during the specified period were included. Outpatients or those admitted for other respiratory diagnoses were excluded;Complete medical records: Patients whose medical records were complete, including laboratory results, clinical notes, and follow-up data, were considered eligible for inclusion.

This cohort represents all pertussis cases diagnosed and treated as inpatients during the observation period at our institution. Any exclusions or missing data were noted and managed according to the study’s ethical guidelines.

Out of the 38 pediatric patients included in the study, 7 were initially reported by caregivers as vaccinated against *Bordetella pertussis*. However, after cross-checking with the National Vaccination Registry via the family physicians, it was confirmed that 4 of these 7 children had incomplete vaccination schedules, having missed one or more of the recommended booster doses. These patients were still classified within the vaccinated group (BPV Yes) but were identified as incompletely vaccinated in our records. The age distribution of the vaccinated patients was as follows: 3 children were between 1 and 3 years of age, 2 were between 4 and 8 years, and 2 were teenagers aged over 9 years. Importantly, none of the patients had received three doses before the age of 1 year, followed by no booster dose after the first year of life. All patients in the unvaccinated group were confirmed to have had no pertussis vaccination (BPV No), which was verified using the national immunization registry.

### 3.2. Molecular Diagnostics

The CFX96 Real-Time PCR Detection System (Bio-Rad, Hercules, CA, USA) was used to detect and quantify Bordetella pertussis DNA. This system was selected for its established reliability in molecular diagnostics and its compatibility with multiplex assay formats, which benefit respiratory pathogen surveillance. The CFX96 system accommodates 96-well plates and supports simultaneous detection of up to six targets, making it suitable for moderate-throughput analysis. DNA was extracted from nasopharyngeal specimens using standardized protocols before PCR amplification. The qPCR reactions were prepared using a master mix containing a DNA template, specific primers, and a fluorescent dye (SYBR Green). The thermal cycling protocol included an initial denaturation step at 95 °C, followed by 40 amplification cycles consisting of denaturation (95 °C), annealing (55–65 °C), and extension (72 °C). Melt curve analysis was performed to verify product specificity. Data acquisition and analysis were conducted using the CFX Maestro 2.3 Software, which allowed for threshold cycle (Ct) determination and relative quantification.

While qPCR is a sensitive and rapid tool for detecting *B. pertussis*, it has limitations, including potential false negatives in late-stage infections and the inability to distinguish between viable and non-viable organisms.

Therefore, results were interpreted alongside clinical and serological findings to improve diagnostic accuracy.

### 3.3. Serological Diagnostics

In addition to molecular diagnostics, serological testing was used in cases where real-time PCR results were negative, but clinical suspicion of *Bordetella pertussis* infection remained high. The serological diagnosis was based on the quantification of anti-*Bordetella pertussis* IgG antibodies. A > 100 IU/mL threshold indicated a recent or current infection, following national diagnostic guidelines and international standards. IgM and IgA antibody levels were not routinely measured, as they are not included in the current diagnostic algorithm at our institution. All serological analyses were performed using enzyme-linked immunosorbent assay (ELISA) kits (Thermo Fisher Scientific Inc., Waltham, MA, USA) approved for clinical use, ensuring diagnostic accuracy. All samples were processed in the hospital’s certified microbiology laboratory.

### 3.4. Imaging and Laboratory Tests

In addition to qPCR, diagnostic imaging (chest X-ray) was performed to evaluate pulmonary involvement. Laboratory tests included serological assays and culture by standard diagnostic practices for pertussis in children.

Typical reference values for the laboratory tests and imaging criteria for interpreting chest X-rays were used based on established local guidelines for pediatric patients.

### 3.5. Statistical Analysis

The extensive data analysis was performed using XLSTAT Life Sciences Software v. 2024.3.0 1423 by Lumivero (Denver, CO, USA). Descriptive statistics analyzed data, and clinical patterns, complications, and outcomes were identified to assess the burden and severity of *B. pertussis* infection in the pediatric population. The variable parameters are displayed as absolute frequencies (number, *n*) and relative frequencies (percentage) [40]. The Fisher Exact Test determined the statistically significant differences at *p* < 0.05 [41]. Using Pearson Correlation, the Principal Component Analysis was performed to investigate the correlations between variable parameters, calculating the correlation coefficient (r) value [42]. The closer r is to 1, the stronger the correlation between variable parameters.

## 4. Discussion

*Bordetella pertussis* infection can affect individuals across all age groups, with a wide range of clinical manifestations posing a significant global public health challenge [43]. The number of reported cases and incidences of pertussis disease are collected annually through the WHO/UNICEF Joint Reporting Form on Immunization [44]. Each country’s news is updated and made publicly available quickly. All recent information results in global trends and leads to evidence-based decisions in different policies (for example, the need for booster vaccine doses and targeted age groups) [45,46]. The present study aims to conduct a complex investigation of the clinical and epidemiological aspects of *Bordetella pertussis* infection in children, focusing on disease burden, risk factors, and vaccination efficacy. It involves 38 pediatric patients from the Southeast region of Romania hospitalized with *B. pertussis* infection in the Pediatric Departments of Constanta County Clinical Emergency Hospital “St. Apostle Andrew” between 1 January 2024 and 30 September 2024, with most of them being up to 3 years of age (31/38, 81.58%).

### 4.1. Clinical Insights

Our study results emphasize the need for comprehensive diagnostic approaches in pediatric patients presenting with respiratory symptoms, as co-infections can complicate the clinical picture and delay effective treatment. Early detection of these pathogens through advanced molecular techniques, such as qPCR, and careful monitoring of respiratory symptom progression can significantly improve patient outcomes and prevent complications associated with these infections.

Our findings highlight the significant prevalence of various pathogens’ co-detection in pediatric patients with respiratory symptoms. Notably, the timing of pathogen detection before presentation to the Emergency Care Unit (ECU) shows a clear correlation with the clinical outcomes and severity of respiratory infections. The highest co-detection frequency (31.58%) occurred between 4–9 days before admission, where *Streptococcus pneumoniae* was the predominant co-pathogen, detected in 50% of these cases. Respiratory viruses, such as enteroviruses (EV-HRV) and SARS-CoV-2, were also frequently co-detected in this period, each found in 25% of patients. Other pathogens, such as *Human Parainfluenza Virus-3* (HPIV-3), *Human Adenovirus* (HAdVs), *Mycoplasma pneumoniae*, and *Pneumocystis jirovecii*, were less commonly co-detected but were still notable, each appearing in 8.33% of the patients. However, while the co-detection of multiple pathogens is observed, it is crucial to note that the mere presence of these pathogens does not necessarily imply a direct causative role in the severity of respiratory illness. Our study highlights that these pathogens may be co-infections rather than active contributors to disease progression. Therefore, further investigation is needed to better understand the interactions between co-detected pathogens and their potential influence on the clinical outcomes of pediatric respiratory diseases [47].

These findings are consistent with the existing literature highlighting the role of co-infection in pediatric respiratory diseases’ severity and complexity [48,49,50,51]. The co-detection of bacterial and viral pathogens is essential in pediatric patients with acute respiratory infections. They influence the clinical progression of pertussis infections and the outcome of pneumonia and bronchiolitis [52]. *S. pneumoniae* has been identified as a frequent co-infecting bacterium in viral respiratory infections, particularly in children with influenza or respiratory viral infections [53]. Similarly, RSV and EV-HRV have been implicated in severe respiratory illness when combined with other respiratory pathogens [54,55].

This study confirms that pertussis manifests with a broad range of clinical symptoms in pediatric patients, from a mild, persistent cough to severe paroxysmal episodes that can lead to complications such as apnea, pneumonia, and even encephalopathy [56]. The disease was most severe in infants under 6 months old, who had the highest hospitalization rates. In severe cases, infants may develop pneumonia and/or respiratory failure, necessitating advanced therapeutic interventions such as conventional mechanical ventilation, high-frequency oscillatory ventilation, plasmapheresis, or extracorporeal membrane oxygenation (ECMO) [57,58,59].

It is clinically significant to distinguish between typical lower respiratory tract infections (LRTIs), such as pneumonia and bronchitis, and acute respiratory distress syndrome (ARDS), which represents a distinct and more severe entity characterized by non-cardiogenic pulmonary edema and refractory hypoxemia [37,60,61]. In our study, ARDS developed in two unvaccinated infants under 1 year of age who had marked hyperleukocytosis and respiratory co-infections. Both required intensive care and mechanical ventilation. This aspect underscores the risk of severe complications in young, unvaccinated children with pertussis, especially in the presence of hyperinflammation or co-infection [62].

The poorest prognosis is linked to hyperleukocytosis with a predominance of lymphocytes, making ECMO a crucial intervention to lower leukocyte counts [63]. It has been suggested that these fatalities result from leukocyte aggregation in small pulmonary vessels, leading to irreversible pulmonary hypertension [64,65]. Several retrospective studies have identified hyperleukocytosis as a significant risk factor for mortality in young infants and an independent predictor of fatal outcomes across all age groups [28,31,66]. Therefore, an elevated white blood cell (WBC) count is a hallmark feature of pertussis. It is significantly higher in severe cases than in mild cases and reaches extreme levels in fatal cases [67,68,69]. The impaired deformability of white blood cells contributes to their tendency to obstruct narrowed alveolar capillaries. Unlike erythrocytes, which pass through these vessels efficiently, leukocytes require 10–15 times longer, leading to embolism due to leukocyte aggregation [62]. This blockage can result in hypoxemia and pulmonary hypertension, which, in turn, can compromise cardiac function and lead to heart failure in critical cases [70,71]. Severe hyperleukocytosis, defined as a WBC count exceeding 50 × 10^3^/µL, has been identified as an independent risk factor for malignant pertussis, a life-threatening form of the disease [33]. Research has demonstrated that in cases where leukocyte counts exceeded 100 × 10⁹/L and no targeted interventions were implemented to reduce WBC levels, standard treatment alone was ineffective, resulting in fatal outcomes in all reported instances [72]. Our findings align with previous research indicating that an elevated white blood cell (WBC) count is a key clinical feature of pertussis, particularly in severe and fatal cases. The data suggest that leukocytosis contributes significantly to disease progression and poor outcomes. Four children experienced complications, including bilateral apical-lateral-basal pneumothorax, a pulmonary condensation process in the lower third of the right pulmonary field, confluent alveolar opacities, and interstitial peri- and hylo-basal bilateral enlarged congestive hilar regions. Additional very severe complications included aspiration pneumonia, kidney failure, paroxysmal manifestations, and anasarca. Our study reports that two pediatric patients did not survive despite comprehensive medical intervention.

Complications associated with pertussis, including post-tussive vomiting, sleep disturbances, and rib and vertebral fractures caused by intense coughing episodes, remain overlooked and under-reported in clinical practice [73]. The impact of long-term respiratory consequences also requires further investigation, as studies have shown that persistent cough and airway hyperreactivity can persist for months following recovery [74,75].

The antibacterial treatment [76], including macrolides [77] and beta-lactams [78], was administered according to international and national guidelines for managing pertussis [79]. Based on clinical indication, the antiviral treatment was mainly used for co-infections with other viral pathogens, such as Remdesivir, in cases involving SARS-CoV-2 [80].

### 4.2. Epidemiological Insights

Pertussis disease is particularly dangerous for infants and young children, as it can cause serious complications (pneumonia, extensive lung lesions, encephalitis, etc.) and even death [81]. It is estimated that the immunity determined by the disease is not permanent, which explains the frequent reinfection of some patients [82]. The vaccine against pertussis was introduced in Romania in 1960, and the routine immunization program for children led to substantial reductions in the occurrence of the disease; 2008 was the year of the switch to aP vaccines for the primary series of vaccination, the reinforcing dose, and preschool booster [83]. Although pertussis is recognized as a vaccine-preventable disease, our findings show that only 18.4% of pediatric patients received the *B. pertussis* vaccine. This aspect is similar to those from other countries regarding a decline in vaccination coverage by 24 months [84]. Moreover, our pediatric patients did not benefit from maternal pertussis immunization; therefore, the highest incidence of whooping cough in infants <1 year (44.7%) is because Pertussis vaccine acceptancy by Romanian pregnant women decreased from 85% in 2019 to 44.4% in 2022 (*p* < 0.01), in association with an increasing pregnancy age and a considerable diminution of the educational level [85]. Of the seven vaccinated pediatric patients, three are young children (1–3 years), and two are 4–8 years old. The other two are teenagers (>9 years). Our findings are similar to those recently reported in Europe [86].

The minimal percentage of vaccinated pediatric patients could result from the worldwide immunization routine regression during the COVID-19 pandemic [87]. The parental anti-vaccination beliefs could also explain the progressive incidence of pertussis [88,89,90,91] and other preventable infectious diseases in childhood [92,93,94,95].

The clinical outcomes, hospitalization period, and therapeutic protocol analysis highlight the benefits of pertussis vaccination in pediatric patients. Hence, the incidence of fever, LRTI, rhinorrhea, and O2 DS is appreciably lower in vaccinated children. The highest WBC, Lym, and CRP levels were found in non-vaccinated pediatric patients. Fewer vaccinated pediatric patients were hospitalized; they had no complications and did not need HFNCO2 and special care in the PICU. Furthermore, the deceased children were unvaccinated. Our results are consistent with previous research, indicating that younger infants face the most significant risk of severe pertussis complications [28] and death caused by their underdeveloped immune systems due to a lack of or incomplete vaccination [5,96].

Our study confirms previous research, indicating that pertussis is diagnosed more often in urban areas than in rural regions. In addition, almost all vaccinated pediatric patients had urban residences. This discrepancy may be due to higher population density, better healthcare access, frequent testing, and greater awareness among urban healthcare professionals [97,98]. However, it does not necessarily imply a lower disease burden in rural areas; instead, it suggests the possibility of under-reporting due to limited diagnostic resources [99].

Regarding pertussis management, the National Institute for Public Health from Romania publicly displayed the following recommendations [35]:Seeing a doctor when symptoms suggestive of the disease appear: intense and prolonged coughing fits, vomiting after coughing fits, difficulty breathing in infants, and noisy inhales (“whooping cough”);Timely vaccination of infants and recovery of arrears;Vaccination of pregnant women to ensure the newborn’s and mother’s protection;Adults’ vaccination, with a booster every 10 years;Promotion of “cocooning” vaccination in the infant’s entourage (in a family expecting a newborn, the parents, grandparents, and siblings of the infant should be vaccinated according to the national vaccination calendar or recommendations for adults).

Although vaccination campaigns are widespread, the continued occurrence of pertussis cases in fully immunized children highlights the shortcomings of current acellular pertussis vaccines [100]. Szwejser-Zawislak et al. found that children who received their last dose more than 5 years ago faced a significantly higher risk of infection, further supporting evidence of waning immunity [101]. This finding strengthens the growing consensus that aP vaccines offer only short-term protection, emphasizing the need for alternative approaches, including (i) administering booster doses earlier and more frequently [102,103] (ii) re-evaluating the use of whole-cell pertussis vaccines in specific populations [101,104,105], and (iii) developing enhanced pertussis vaccines with longer-lasting immunity [106,107,108]. Recent advancements in the development of live attenuated pertussis vaccines (LAVs) have shown encouraging results in preclinical research, offering stronger mucosal immunity and more durable protection than acellular vaccines [109,110,111,112]. These findings underscore the importance of continued vaccine innovation in overcoming existing pertussis prevention and control challenges [107,113,114,115,116]. Moreover, vaccinating mothers during pregnancy has been proven to lower the risk of severe pertussis in young infants by providing passive immunity [117,118,119].

### 4.3. Limitations

Despite its valuable findings, this study has several limitations that should be mentioned.

As a retrospective analysis, the study relies on previously recorded medical data, which may introduce biases such as incomplete documentation, variability in clinical assessments, and potential misclassification of diagnoses.The qPCR is a powerful tool for diagnosing respiratory infections, but it has limitations. It detects both viable and non-viable pathogens, which can lead to overdiagnosis and misinterpretation of clinical significance. It does not provide information on pathogen load, viability, or the host immune response, making it insufficient for determining disease severity or guiding treatment. Additionally, qPCR cannot detect emerging pathogens or strains not included in specific panels, and false positives or contamination can occur. For a more accurate diagnosis, qPCR results should be interpreted alongside clinical symptoms, patient history, and additional diagnostic tests.The study included only 38 pediatric patients, which may not represent the broader population of children affected by *Bordetella pertussis* infection. The small sample size may limit the generalizability of the findings and reduce statistical power for specific associations.The data were collected from a single hospital (Constanta County Clinical Emergency Hospital St. Apostle Andrew), which may not fully capture regional or national variations in pertussis infection patterns, co-infections, and clinical management strategies. Future studies should incorporate a prospective, multicenter design with a larger cohort, include long-term follow-up of patients, and evaluate the impact of vaccination status and socioeconomic factors on disease outcomes.

### 4.4. Essential Considerations

Our study revealed that many cases were initially mistaken for viral respiratory infections, resulting in delays in appropriate treatment and contact tracing. While qPCR testing has enhanced early detection, its sensitivity diminishes as the disease progresses, particularly in later stages. The accurate and timely diagnosis of pertussis remains a significant challenge. Improving clinical awareness and expanding laboratory capacity are crucial for enhancing detection rates and controlling transmission. Serological testing is increasingly being investigated as a supplementary diagnostic method, especially for young children with persistent coughs [120,121,122]. However, the widespread adoption of this approach remains limited due to the lack of standardized serological assays [123].

Another major challenge is the under-reporting and misclassifying of pertussis cases in national surveillance systems, contributing to underestimating the disease burden. Strengthening integrated surveillance systems, raising physician awareness, and improving laboratory testing capabilities are essential to address these issues [35]. The variation in positivity rates may be linked to the underdiagnosis of *Bordetella pertussis*, either due to the lack of routine testing for the bacterium or, particularly during the winter season, the potential misinterpretation of symptoms as those of other common respiratory infections [52,124].

Multiple factors, including maternal vaccination during pregnancy and the child’s age, influence the frequency and severity of pertussis in infants [125,126,127]. The clinical severity observed in the two fatal cases underscores the high risk associated with pertussis in young, unvaccinated infants, a population too young to have completed primary immunization. Severe hyperleukocytosis, as noted in one of these cases, has been associated with poor outcomes and may reflect immune dysregulation or leukostasis, contributing to pulmonary hypertension and multi-organ failure. Our findings align with these reports and reinforce the urgency of timely vaccination and the exploration of adjunctive therapies—such as leukoreduction strategies—in hyperleukocytic patients [66]. Even when initiated, incomplete immunization may not confer sufficient protection, particularly in the face of waning immunity and delayed boosters [110]. However, a significant factor sustaining disease transmission could be asymptomatic carriage and underdiagnosis, especially among older children and adults, who act as reservoirs for infection, as previous findings reported [124,128]

Several studies have shown that the immunity conferred by the acellular pertussis vaccine wanes more rapidly than expected, particularly in the absence of timely booster doses. It is especially relevant for older children and adolescents who, despite completing the primary vaccination series in early childhood, may become susceptible again to *Bordetella pertussis* infection within a few years [110,129]. Furthermore, while parental reporting may suggest complete vaccination, verification via the national immunization registry revealed that some children had incomplete vaccination histories, underlining the importance of accurate vaccination documentation in clinical assessments. This waning immunity poses a significant public health concern, as it contributes to the re-emergence of pertussis in populations with suboptimal booster coverage. Studies have also highlighted that natural infection and vaccination provide limited long-term protection, reinforcing the need for updated immunization strategies [130,131,132,133].

## 5. Conclusions

Despite widespread vaccination campaigns, *Bordetella pertussis* remains a significant public health concern, primarily due to waning immunity and diagnostic limitations. Combatting anti-vaccination practices through solid and accurate medical education, strengthening booster vaccination programs, improving early detection, and enhancing surveillance efforts are critical for effective pertussis control. Bridging these gaps through ongoing research and targeted public health initiatives will be essential in reducing the disease burden among children and ensuring more sustained protection against pertussis.

## Figures and Tables

**Figure 1 antibiotics-14-00428-f001:**
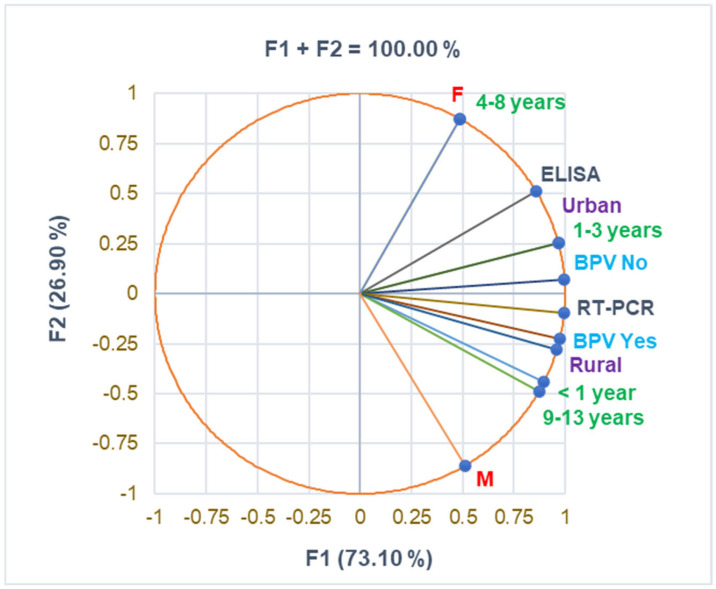
The correlations between baseline variable parameters. BPV = *Bordetella pertussis* vaccine; F—female; M—male; RT-PCR—reverse transcription-polymerase chain reaction; ELISA—serological diagnosis.

**Figure 2 antibiotics-14-00428-f002:**
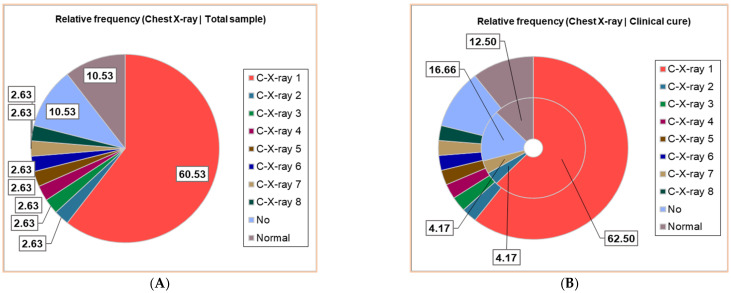
The correspondence between radiological findings and illness severity: (**A**). Total sample; (**B**). Clinical cure; (**C**). Death; (**D**). Transfer. C-X-ray 1—Accentuated interstitial pattern below the hilum bilaterally; C-X-ray 2—Alveolar opacities around and below the right hilum; C-X-ray 3—Bilateral pulmonary infiltrate; C-X-ray 4—Blurring of the right basal pulmonary field: C-X-ray 5—Congestive pulmonary hila, microalveolar, and reticular opacities around and below the hilum bilaterally; C-X-ray 6—Enlarged congestive hila, confluent alveolar opacities around the left hilum and below the hilum bilaterally; C-X-ray 7—Right pulmonary consolidation process; C-X-ray 8—Widespread, homogeneous opacities of medium intensity, with blurred margins, showing air bronchograms, located in the upper third of both lung fields and left retrocardiac area—indicative of pulmonary infiltrates. All data presented were obtained using Descriptive Statistics, available in extensive form in Appendix A.

**Figure 3 antibiotics-14-00428-f003:**
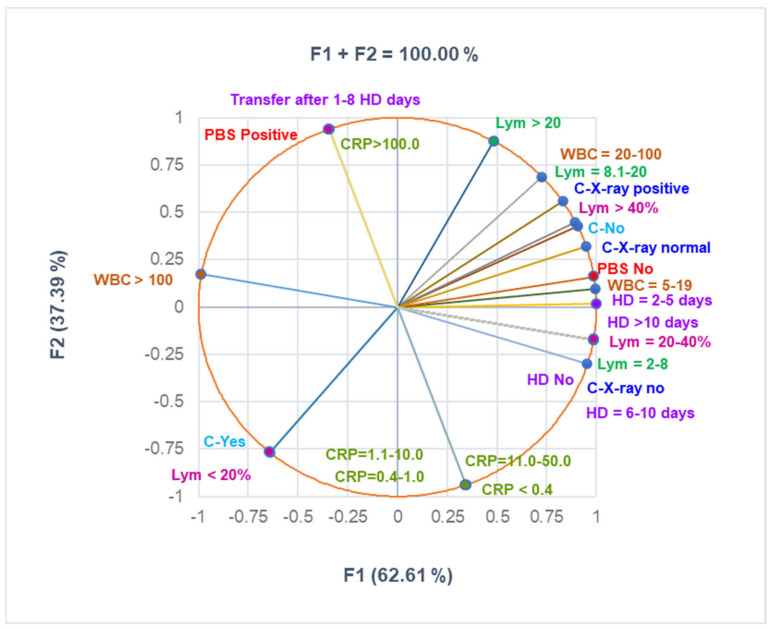
The correlations between clinical laboratory analyses, radiological examination, complications, and hospitalization period. PBS—peripheral blood smear—all PBS abnormalities are detailed in Appendix A. CRP—C-reactive protein: normal: <0.4 mg/mL; moderately increased: 0.4–1.0 mg/mL; high level: 1.1–10 mg/mL; C-complications; WBCs—White blood cells; Lym—Lymphocytes; C-X-ray—Chest radiography; HD—Hospitalization days.

**Figure 4 antibiotics-14-00428-f004:**
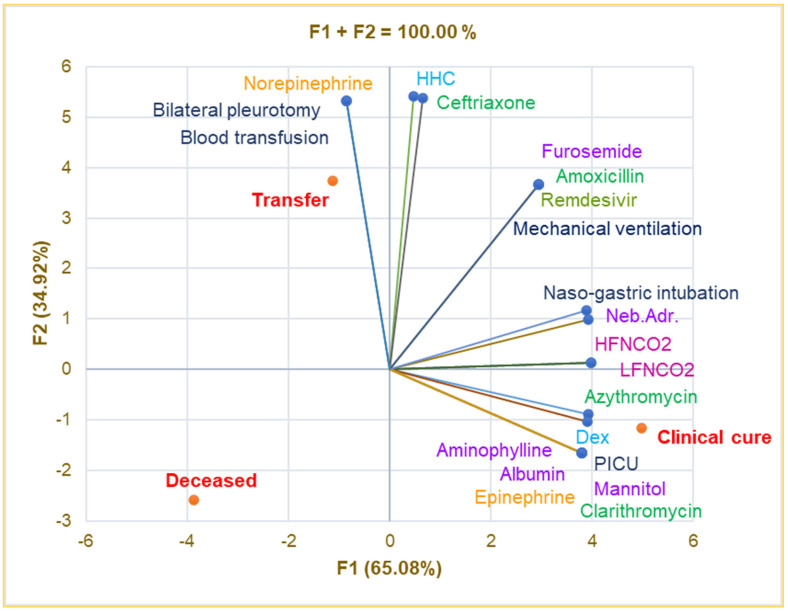
Correlations between therapy and clinical outcomes in pediatric patients. PICU—Pediatric intensive care unit; Dex—Dexamethasone; HHC—Hydrocortisone hemisuccinate; Neb.Adr.—Nebulized adrenaline; HFNCO2—High-flow nasal cannula oxygen therapy; LFNCO2—Low-flow nasal cannula oxygen therapy.

**Figure 5 antibiotics-14-00428-f005:**
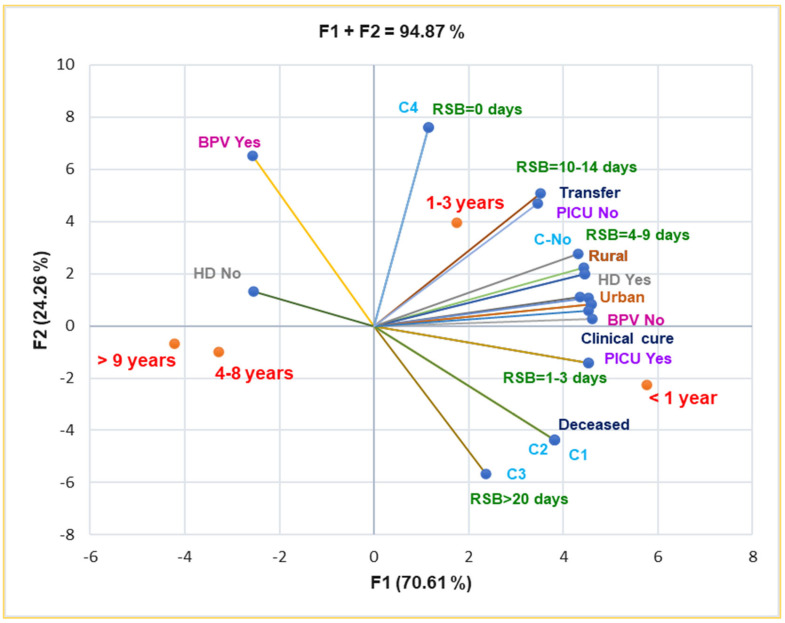
Correlations between baseline data and clinical outcomes in pediatric patients. RSB—Respiratory symptoms before presentation in ECU; HD—Hospitalization days, C–Complications; PICU—Pediatric intensive care unit; BPV—*B. pertussis* vaccination.

**Table 1 antibiotics-14-00428-t001:** Baseline data of the pediatric patients’ group (*n* = 38).

Aspect		Total		F		M		*p*-Value
		*n*	*%*	*n*	*%*	*n*	*%*	
Total		38	100	19	50	19	50	
Age (years)	<1 year	17	44.74	6	31.58	11	57.89	<0.05 <0.05
	1–3 years	14	36.84	8	42.11	6	31.58	<0.05
	4–8 years	4	10.53	4	21.05	0	0.00	<0.05
	9–13 years	3	7.89	1	5.26	2	10.53	<0.05
Residence	Rural	17	44.74	7	36.84	10	52.63	<0.05
	Urban	21	55.26	12	63.16	9	47.37	<0.05
*B. pertussis* vaccination	No	31	81.58	16	84.21	15	78.95	>0.05
	Yes	7	18.42	3	15.79	4	21.05	<0.05
*B. pertussis* diagnosis	ELISA	6	15.79	4	21.05	2	10.53	<0.05
	qPCR	32	84.21	15	78.95	17	89.47	>0.05

*n* = number (frequency), and % = relative frequency; *p* < 0.05 reports statistically significant differences.

**Table 2 antibiotics-14-00428-t002:** Co-detected pathogens in the case of respiratory symptoms claimed before presentation (RSB) in the Emergency Care Unit (days).

Aspect	Total		Respiratory Symptoms Claimed Before Presentation in ECU (Days)									
			No		1–3 Days		4–9 Days		10–14 Days		>20 Days	
	*n*	%	*n*	%	*n*	%	*n*	%	*n*	%	*n*	%
Total	38	100	2	5.26	7	18.42	12	31.58	12	31.58	5	13.16
Viruses												
EV-HRV	12	31.58	0	0.00	3	42.86	3	25.00	3	25.00	3	60.00
HPIV-3	2	5.26	0	0.00	1	14.29	1	8.33	0	0.00	0	0.00
SARS-CoV2	5	13.16	0	0.00	2	28.57	3	25.00	0	0.00	0	0.00
HAdVs	1	2.63	0	0.00	0	0.00	1	8.33	0	0.00	0	0.00
MV	1	2.63	0	0.00	0	0.00	0	0.00	0	0.00	1	20.00
RSV	1	2.63	0	0.00	1	14.29	0	0.00	0	0.00	0	0.00
Other Pathogens												
*S. pneumoniae*	7	18.42	0	0.00	1	14.29	6.00	50.00	0	0.00	0	0.00
*M. pneumoniae*	1	2.63	0	0.00	0	0.00	1.00	8.33	0	0.00	0	0.00
*P. jirovecii*	1	2.63	0	0.00	0	0.00	1.00	8.33	0	0.00	0	0.00

ECU—Emergency Care Unit; EV-HRV—Enterovirus-human-rhinovirus; SARS-CoV-2—Coronavirus type 2; HPIV-3—Human parainfluenza virus type 3; HAdVs—Human adenovirus; MV—Measles virus (Morbillivirus); RSV—Respiratory syncytial virus; *n* = number (frequency), and % = relative frequency.

**Table 3 antibiotics-14-00428-t003:** Clinical laboratory analyses, radiological examination, complications, and hospitalization period.

Parameter		Total		Outcome					
				Clinical Cure		Deceased		Transfer	
		*n*	*%*	*n*	*%*	*n*	*%*	*n*	*%*
Laboratory Analyses									
WBC *n* × 10^3^	WBC = 20–100	18	47.37	9	37.50	1	50.00	8	66.67
	WBC = 5–19	18	47.37	15	62.50	0	0.00	3	25.00
	WBC > 100	2	5.26	0	0.00	1	50.00	1	8.33
Lym%	Lym < 20%	1	2.63	0	0.00	1	50.00	0	0.00
	Lym = 20–40%	7	18.42	5	20.83	1	50.00	1	8.33
	Lym > 40%	30	78.95	19	79.17	0	0.00	11	91.67
Lym *n* × 10^3^	Lym = 2–8	14	36.84	12	50.00	1	50.00	1	8.33
	Lym = 8.1–20	12	31.58	7	29.17	0	0.00	5	41.67
	Lym > 20	12	31.58	5	20.83	1	50.00	6	50.00
PBS	No	31	81.58	23	95.83	1	50.00	7	58.33
	PBS positive	7	18.42	1	4.17	1	50.00	5	41.67
CRP mg/dL	<0.4	13	34.21	9	37.50	0	0.00	4	33.33
	0.4–1.0	8	21.05	6	25.00	0	0.00	2	16.67
	1.1–10.0	11	28.95	7	29.17	0	0.00	4	33.33
	11.0–50.0	4	10.53	2	8.33	1	50.00	1	8.33
	51.0–100.0	1	2.63	0	0.00	1	50.00	0	0.00
	>100.0	1	2.63	0	0.00	0	0.00	1	8.33
Chest X-ray									
N/A		4	10.53	4	16.67	0	0.00	0	0.00
Normal		4	10.53	3	12.50	0	0.00	1	8.33
Positive		30	78.95	17	70.83	2	100.00	11	91.67
Complications									
C-No		34	89.47	23	95.83	0	0.00	11	91.67
C-Yes		4	10.53	1	4.17	2	100.00	1	8.33
Days of Hospitalization									
2–5 days		4	10.53	4	16.66	0	0.00	0	0.00
6–10 days		16	42.11	14	58.33	2	100.00	0	0.00
>10 days		2	5.26	2	8.33	0	0.00	0	0.00
No		4	10.53	4	16.66	0	0.00	0	0.00
Transfer after 1–8 days		12	31.58	0	0.00	0	0.00	12	100.00

PBS—peripheral blood smear—all PBS abnormalities are detailed in Appendix A. CRP—C reactive protein: normal: <0.4 mg/mL; moderately increased: 0.4–1.0 mg/mL; high level: 1.1–10 mg/mL; C-complications; WBCs—white blood cells—They were measured by the number of white blood cells per cubic millimeter of blood (cells/mm^3^); normal WBC counts by age are as follows: Babies 0 to 2 weeks old: 9.000 to 30.000 cells/mm^3^; babies 2 to 8 weeks old: 5.000 to 21.000 cells/mm^3^; children 2 months to 6 years old: 5.000 to 19.000 cells/mm3; children 6 to 18 years old: 4.800—10.800 cells/mm^3^ [38]. Lym—Lymphocytes are measured in cells per microliter of blood (cells/µL). Normal levels: children: 3.000–9.500 cells/µL. Lymphocytosis is defined as a lymphocyte count of more than 8.000 cells/mL (for young children) and over 4.000 cells/µL (for teenagers and adults) [39]. The percentage of lymphocytes in the total WBC population declines with age from 67.4% of the WBC in infants to 47.5% in young children to 38.2% in older children. Chest X-ray—Chest radiography.

**Table 4 antibiotics-14-00428-t004:** Treatment and evolution of pediatric patients.

Aspect	Total		Clinical Cure		Deceased		Transfer	
	*n*	%	*n*	%	*n*	%	*n*	%
Total	38	100	24	63.16	2	5.26	12	31.58
PICU	7	18.42	5	20.83	1	50.00	1	8.33
Antimicrobial Drugs								
Amoxicillin	2	5.26	1	4.17	0	0.00	1	8.33
Azithromycin	21	55.26	15	62.50	2	100.00	4	33.33
Ceftriaxone	8	21.05	2	8.33	0	0.00	6	50.00
Clarithromycin	5	13.16	5	20.83	0	0.00	0	0.00
Remdesivir	2	5.26	1	4.17	0	0.00	1	8.33
Other Drugs								
Dex	21	55.26	17	70.83	1	50.00	3	25.00
HHC	14	36.84	4	16.67	1	50.00	9	75.00
Neb.Adr.	31	81.58	19	79.17	2	100.00	10	83.33
Aminophylline	1	2.63	1	4.17	0	0.00	0	0.00
Epinephrine	1	2.63	1	4.17	0	0.00	0	0.00
Norepinephrine	1	2.63	0	0.00	0	0.00	1	8.33
Furosemide	2	5.26	1	4.17	0	0.00	1	8.33
Mannitol	1	2.63	1	4.17	0	0.00	0	0.00
Albumin	1	2.63	1	4.17	0	0.00	0	0.00
Other Therapeutical Interventions								
HFNCO2	4	10.53	3	12.50	0	0.00	1	8.33
LFNCO2	18	47.37	11	45.83	2	100.00	5	41.67
Nasogastric Intubation	3	7.89	2	8.33	0	0.00	1	8.33
Mechanical ventilation	2	5.26	1	4.17	0	0.00	1	8.33
Bilateral pleurotomy	1	2.63	0	0.00	0	0.00	1	2.63
Blood transfusion	2	5.26	0	0.00	0	0.00	2	16.67

PICU—Pediatric intensive care unit; Dex—Dexamethasone, HHC—Hydrocortisone hemisuccinate; Neb.Adr.—Nebulized adrenaline; HFNCO2—High-flow nasal cannula O_2_ therapy; LFNCO2—Low-flow nasal cannula O_2_ therapy.

**Table 5 antibiotics-14-00428-t005:** The most significant baseline and clinical insights associated with BP vaccination status.

Parameter	BPV No		BPV Yes		*p*-Value
	*n*	*%*	*n*	*%*	
Total	31	81.58	7	18.42	<0.05
Residence					
Rural	16	51.61	1	14.29	<0.05
Urban	15	48.39	6	85.71	<0.05
Age (years)					
<1 year	17	54.84	0	0.00	<0.05
1–3 years	11	35.48	3	42.86	<0.05
4–8 years	2	6.45	2	28.57	<0.05
>9 years	1	3.23	2	28.57	<0.05
Clinical Manifestations					
Fever	14	45.16	2	28.57	<0.05
LRTI	21	67.74	2	28.57	<0.05
Rhinorrhea	13	41.94	1	14.29	<0.05
O2 DS	7	22.58	0	0.00	<0.05
Diagnostic					
C-X-ray-Normal	3	9.68	1	14.29	<0.05
PBS No	24	77.42	7	100.00	<0.05
WBC (*n* × 10^3^/µL)					
WBC = 20–100	15	48.39	3	42.86	>0.05
WBC > 100	2	6.45	0	0.00	<0.05
Lym (*n* × 10^3^/µL)					
Lym = 8.1–20	10	32.26	2	28.57	>0.05
Lym > 20	11	35.48	1	14.29	<0.05
CRP (mg/dL)					
0.4–1.0	5	16.13	3	42.86	<0.05
1.1–10.0	10	32.26	1	14.29	<0.05
51.0–100.0	1	3.23	0	0.00	<0.05
>100.0	1	3.23	0	0.00	<0.05
Hospitalization					
HS-Yes	30	96.77	4	57.14	<0.05
HD = 6–10 days	14	45.16	2	28.57	<0.05
Oxygen Administration					
HFNCO2	4	12.90	0	0.00	<0.05
LFNCO2	16	51.61	2	28.57	<0.05
Outcomes					
C-No	27	87.10	0	100.00	<0.05
PICU Yes	7	22.58	0	0.00	<0.05
Deceased	2	6.45	0	0.00	<0.05

PBS—peripheral blood smear—all PBS abnormalities are detailed in Appendix A. O2 DS—Oxygen desaturation. CRP—C reactive protein: normal: <0.4 mg/mL; moderately increased: 0.4–1.0 mg/mL; high level: 1.1–10 mg/mL. WBCs—White blood cells; HFNCO2—High-flow nasal cannula O2 therapy; LFNCO2—Low-flow nasal cannula O2 therapy; C-complications; PICU—Pediatric intensive care unit; LRTI—Low respiratory tract infection.

## Data Availability

Due to the ongoing study, the data presented in this study are available upon request from the first author and the first corresponding author.

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
