# Peer review of "Clinical and Epidemiological Characteristics of Pediatric Pertussis Cases: A Retrospective Study from Southeast Romania"

_antibiotics, 2025, doi:10.3390/antibiotics14050428_

Round 1
Reviewer 1 Report
Comments and Suggestions for Authors
Unfortunately, this article cannot fulfill the expectations raised by the title and abstract. It is a retrospective case series of children with pertussis, which is quite interesting in itself. Equally interesting are the additional insights into epidemiology, clinical presentation, pathophysiology and vaccination prevention. One of the major problems with this article is the almost meaningless application of complex statistical test procedures to a small number of cases. In this way, “correlations” are described that may simply be based on chance. Complex data concerning the detection of multiple other viruses and bacteria (pathogens?) using multiplex PCR require critical evaluation. For example, the detection of rhinovirus or S. pneumoniae by PCR in respiratory samples is not sufficient to explain any influence on the clinical severity of disease in pertussis. Pertussis in an unvaccinated child is undoubtedly one of the important differential diagnoses of RSV bronchiolitis in the first 6 months of life. Do these 38 patients represent all cases detected and treated as inpatients in the obersevation period? The description of the molecular diagnostic tool sounds as copypasted from the manufacturer's sales brochure. It is unclear, which serologic test results have confirmed Pertussis in the 6 children without positive PCR (IgG treshold? IgM? IgA?) ARDS (acute respiratory distress syndrome) is not synonymous to "lower respiratory tract infection" I suggest to focus on the clincial dataset (shortened and more focused; more detailed description of the two deceased children) and the results of the literature research.
Comments on the Quality of English LanguageNot detailed.
Author Response
Please, see the attachment.

Reviewer 2 Report
Comments and Suggestions for Authors
Pertussis is a respiratory bacterial infection that can cause severe respiratory distress in all age groups. In the younger children, pertussis can be a life-threaten disease. In this study, the authors describe the frequency of pertussis in different pediatric groups (vaccinated and no- vaccinated), the severity of symptoms and the treatment. The topic of the study is interested in the scientific community. However, the analysis of the results and the overall presentation have to be improved. Herein are some comments for the authors:
Major comment
- The appropriate statistical analysis for the independent variables is the Fisher's exact test. Why the authors use the ANOVA test? The numbers of the patients is not large and there are independent variables.
- Are there children that they have received one of dose of vaccination, but not all the booster doses that are recommended? If yes, in which group (vaccinated or no vaccinated children) these children were categorized?
- Were there any B. pertussis infected children that they received three doses before the age of 1 year old and they did not receive the booster dose after the first year? The authors have to comment that the antibodies against B.pertussis can be dropped dramatically in vaccinated children after the 1 year of life.
Minor comments
- Lines 123 – 126. “Given the rise..” Please, make the long sentence into short sentences. This long sentence is difficult to be read and understood.
- Line 127 Why the 33% of the infants is not eligible for vaccination? Please, explain.
- Line 181-183. Please, rephrase. ANOVA is a statistical test and not a statistical software. Did the authors use just ANONA test or additional tests, as well?
- In table 1, remove the decimal numbers from column N.
Author Response
Pertussis is a respiratory bacterial infection that can cause severe respiratory distress in all age groups. In the younger children, pertussis can be a life-threaten disease. In this study, the authors describe the frequency of pertussis in different pediatric groups (vaccinated and no- vaccinated), the severity of symptoms and the treatment. The topic of the study is interested in the scientific community. However, the analysis of the results and the overall presentation have to be improved. Herein are some comments for the authors:
Dear Reviewer 2,
Thank you for your valuable feedback and for recognizing the relevance of our study topic within the scientific community. We appreciate your suggestions and agree that both the analysis and presentation of the results can be strengthened. In response to your comments, we have revised the manuscript to improve the clarity of the data analysis and enhance the overall structure of the presentation. We have addressed your specific comments below (please see our detailed point-by-point responses, and marked paragraphs in the revised MS), and we hope the revised manuscript meets the standards expected for publication. We are grateful for your insightful review, which has contributed significantly to improving the quality of our work.
Major comment
1. The appropriate statistical analysis for the independent variables is the Fisher's exact test. Why the authors use the ANOVA test? The numbers of the patients is not large and there are independent variables.
R1: Thank you so much for your accurate comment. We revised this aspect using Fisher’s exact test, as suggested.
2. Are there children that they have received one of dose of vaccination, but not all the booster doses that are recommended? If yes, in which group (vaccinated or no vaccinated children) these children were categorized?
R2: Yes, in our cohort, some children had received only part of the recommended vaccination schedule—specifically, one or more doses without completing all required boosters. These patients were classified in the vaccinated group, but with a clear note indicating their incomplete immunization status. This distinction is essential, as partial vaccination may confer limited protection and does not provide the same level of immunity as the full vaccine schedule. We have updated the Methods and Results sections to reflect this classification and clarify how these cases were identified through verification with the national immunization registry: “Out of the 38 pediatric patients included in the study, 7 were initially reported by caregivers as vaccinated against Bordetella pertussis. However, after cross-checking with the National Vaccination Registry via the family physicians, it was confirmed that 4 of these 7 children had incomplete vaccination schedules, having missed one or more of the recommended booster doses. These patients were still classified within the vaccinated group, but were identified as incompletely vaccinated in our records. The age distribution of the vaccinated patients was as follows: 3 children were between 1–3 years of age, 2 were between 4–8 years, and 2 were adolescents aged over 9 years. Importantly, none of the patients had received three doses before the age of 1 year followed by no booster dose after the first year of life. All patients in the unvaccinated group were confirmed to have had no pertussis vaccination at all, which was verified using the national immunization registry.”
3. Were there any B. pertussis infected children that they received three doses before the age of 1 year old and they did not receive the booster dose after the first year? The authors have to comment that the antibodies against B.pertussis can be dropped dramatically in vaccinated children after the 1 year of life.
R3: Thank you for this important question. None of the patients in our cohort had received all three primary doses before 1 year of age and subsequently missed the booster. However, among the 7 pediatric patients classified as vaccinated, 4 were found—after verification via the National Vaccination Registry—to have received incomplete vaccination schedules. Although parents initially claimed full vaccination, this was contradicted by official immunization records accessed through their family physicians.
We classified these children in the “vaccinated” group for consistency with similar studies but we clearly noted their incomplete status in the text. We have updated the Methods section accordingly to explain how vaccination status was verified and how these patients were classified: “Vaccination status was initially recorded based on caregiver report at admission. However, in cases of uncertainty or inconsistency, the vaccination records were verified through the National Vaccination Registry, accessed via the patients’ family physicians. As a result, among the 7 patients initially considered vaccinated, 4 were found to have incomplete vaccination schedules, having missed one or more recommended doses. These patients were classified in the "vaccinated" group but marked as incompletely vaccinated to reflect real-world documentation challenges and for the purpose of stratified analysis.”
Regarding waning immunity, we have expanded the Discussion to emphasize that protection following primary immunization may decline significantly over time if booster doses are not administered: “Several studies have shown that the immunity conferred by the acellular pertussis vaccine wanes more rapidly than previously expected, particularly in the absence of timely booster doses. This is especially relevant for older children and adolescents who, despite having completed the primary vaccination series in early childhood, may become susceptible again to Bordetella pertussis infection within a few years. Furthermore, while parental reporting may suggest complete vaccination, verification via the national immunization registry revealed that some children had incomplete vaccination histories, underlining the importance of accurate vaccination documentation in clinical assessments. This waning immunity poses a significant public health concern, as it contributes to the re-emergence of pertussis in populations with suboptimal booster coverage. Studies have also highlighted that both natural infection and vaccination provide limited long-term protection, reinforcing the need for updated immunization strategies.
Minor comments
1. Lines 123 – 126. “Given the rise..” Please, make the long sentence into short sentences. This long sentence is difficult to be read and understood.
R1: We thank the reviewer for this valuable suggestion. To improve clarity and readability, the originally long sentence has been revised into shorter, more concise statements in accordance with the reviewer’s request. The revised text now reads as follows:
“There has been a rise in whooping cough incidence in the EU/EEA. In Romania, 112 cases of whooping cough were recorded in the first four months of 2024. This marks a significant increase compared to only 16 cases reported in 2023. It is also higher than the average of 93 cases reported annually during the five pre-pandemic years (2015–2019) [34].”
2. Line 127 Why the 33% of the infants is not eligible for vaccination? Please, explain.
R2: We appreciate the reviewer’s comment and will clarify the eligibility for vaccination. In the sentence you referenced, 33% of infants who were not eligible for vaccination were too young to receive the recommended doses of the pertussis vaccine. According to our National vaccination guidelines, the first dose of the pertussis vaccine is typically administered at 2 months of age. Therefore, infants younger than 2 months would not yet be eligible for vaccination. These infants represent a vulnerable group who are at a higher risk of severe disease due to the absence of immunity from vaccination. We will revise the text in the manuscript to provide this clarification for better understanding. The revised sentence would read: “The cases belong to 22 Romanian counties, with the most affected age group being 0-4 years. Of these cases, 33% were infants, of whom 89% were eligible for vaccination. The remaining 11% of infants were too young to receive the pertussis vaccine, as they were under 2 months of age. “
3. Line 181-183. Please, rephrase. ANOVA is a statistical test and not a statistical software. Did the authors use just ANONA test or additional tests, as well?
R3: Thank you for the valuable comment. We used Descriptive Statistics, Fisher's Exact Test, and Principal Component Analysis from XLSTAT Software. https://www.xlstat.com/en/solutions/life-sciences
We rephrased the entire subsection to avoid confusion, as follows (lines 214-221):
The extensive data analysis was performed using XLSTAT Life Sciences Software v. 2024.3.0 1423 by Lumivero (Denver, CO, USA). Descriptive statistics analyzed data, and clinical patterns, complications, and outcomes were identified to assess the burden and severity of B. pertussis infection in the pediatric population. The variable parameters are displayed as absolute frequencies (number, n) and relative frequencies (percentage) [37]. The Fisher Exact Test determined the statistically significant differences, while the Pearson Correlation and Principal Component Analysis investigated the correlations between variable parameters [38]. Statistical significance was established at p < 0.05 [39].
4. In table 1, remove the decimal numbers from column N.
R4: Thank you for the feedback. We revised all tables, including Table 1, to remove the decimal numbers in column n, as requested. The table will be adjusted to display only whole numbers (integers) for the frequency (n) values.
Round 2
Reviewer 1 Report
Comments and Suggestions for Authors
Many thanks for the changes made in the original manuscript, which desribes 38 clinically relevant and interesting cases of pertussis in children treated as inpatients including two with fatal outcome.
Author Response
Dear Reviewer 1,
The authors highly appreciate your excellent and accurate review report in round 1. Moreover, they are grateful for your time and attention in analyzing the revised version and for your excellent appreciation—a precious reward for their efforts to revise the manuscript.
Reviewer 2 Report
Comments and Suggestions for Authors
After the implementation of reviewers’ comments, the manuscript has been significantly improved. However, there are certain parts in the manuscript that have to be clarified or refined. Herein are some additional comments for the authors.
- Lines 224 At least one dose of vaccination? Please, precise.
- In table 1, it is not clear between which groups the statistical analysis was conducted. Is it between the different age groups or between the sex percentages?
- Figures 1 to 5 require further explanation for the presented data. The authors must communicate with more details the information in the figures or the figures have to be presented in another format.
- Lines 576- 586. This is information regarding the methodology of the study and it does not belong in discussion part.
- General comments for the discussion part: The authors have to comment the results of their study and not to present results, e.g. table 5. The authors can emphasize where there is statistical significance and interpreter the results, but they have to delete percentages or numbers.
Author Response
After the implementation of reviewers’ comments, the manuscript has been significantly improved.
The authors are grateful for the Reviewer 2 appreciation, as a reward for their efforts to revise the manuscript according to all review reports.
However, there are certain parts in the manuscript that have to be clarified or refined. Herein are some additional comments for the authors.
The authors are grateful for the Reviewer 2 time, patience, accuracy, high professionalism, and valuable comments aiming to increase the quality of the present MS.
- Lines 224 At least one dose of vaccination? Please, precise.
Response 1: The authors thank Reviewer 2 for this essential comment. They included this significant mention in the MS text in line 232.
2. In table 1, it is not clear between which groups the statistical analysis was conducted. Is it between the different age groups or between the sex percentages?
Response 2: The authors are grateful to Reviewer 2 for this accurate comment. To better understand, they rectified Table 1 and put the p-value in each row.
3. Figures 1 to 5 require further explanation for the presented data. The authors must communicate with more details the information in the figures or the figures have to be presented in another format.
Response 3: The authors thank the Reviewer 2 for such accurate and attentive comments.
Figures 1, 3-5 resulted from Principal Component Analysis through Pearson Correlation, while Figure 2 used Descriptive Statistics with qualitative data. https://www.xlstat.com/en/solutions/features/descriptive-statistics-including-box-plots-and-scattergrams.
Principal component Analysis is a powerful statistical tool that supports the results presented, showing the statistical significance and strength of the correlation between 2 variable parameters. The closer r is to 1, the stronger the correlation between variable parameters. https://www.xlstat.com/en/solutions/features/principal-component-analysis-pca.
They specified it in lines 226-228:
Using Pearson Correlation, Principal Component Analysis investigated the correlations between variable parameters, calculating the correlation coefficient (r) value [38]. The closer r is to 1, the stronger the correlation between variable parameters.
The Supplementary Material detailed the corresponding Statistical Analyses for all Figures in Excel form.
Moreover, all Figures extend the presentation of data registered in tables 1-5. Each Figure is suitably introduced in the MS text – all mentions are marked with yellow. The statistically significant correlations confirm our findings, underline the discussions and support the conclusions.
4. Lines 576- 586. This is information regarding the methodology of the study and it does not belong in discussion part.
Response 4: The authors are grateful for this valuable comment. Table 5 and the entire paragraph with detailed data was transfered in Materials and Methods, lines 162-173.
5. General comments for the discussion part: The authors have to comment the results of their study and not to present results, e.g. table 5. The authors can emphasize where there is statistical significance and interpreter the results, but they have to delete percentages or numbers.
Response 5: The authors are grateful for this accurate comment. Table 5, with all data presentation, was transferred to the Results section (lines 442-464), and the most important findings were discussed in lines 611-622:
The clinical outcomes, hospitalization period, and therapeutic protocol analysis highlight the benefits of pertussis vaccination in pediatric patients. Hence, the incidence of fever, LRTI, rhinorrhea, and O2 DS is appreciably lower in vaccinated children. The highest WBC, Lym, and CRP levels were found in non-vaccinated pediatric patients Less numerous vaccinated pediatric patients were hospitalized; they had no complications, and did not need HFNCO2 and special care in PICU. Furthermore, the deceased children were unvaccinated. Our results are consistent with previous research, indicating that younger infants face the most significant risk of severe pertussis complications [28] and death caused by their underdeveloped immune systems due to a lack of or incomplete vaccination [5,96].
Our study confirms previous research, indicating that pertussis is diagnosed more often in urban areas than rural regions. In addition, almost all vaccinated pediatric patients had urban residences.